# Variable Correlation between Bronchoalveolar Lavage Fluid Fungal Load and Serum-(1,3)-β-d-Glucan in Patients with Pneumocystosis—A Multicenter ECMM Excellence Center Study

**DOI:** 10.3390/jof6040327

**Published:** 2020-12-01

**Authors:** Toine Mercier, Nesrine Aissaoui, Maud Gits-Muselli, Samia Hamane, Juergen Prattes, Harald H. Kessler, Ivana Mareković, Sanja Pleško, Jörg Steinmann, Ulrike Scharmann, Johan Maertens, Katrien Lagrou, Blandine Denis, Stéphane Bretagne, Alexandre Alanio

**Affiliations:** 1Department of Microbiology, Immunology and Transplantation, KU Leuven, 3000 Leuven, Belgium; toine.mercier@uzleuven.be (T.M.); johan.maertens@uzleuven.be (J.M.); katrien.lagrou@uzleuven.be (K.L.); 2Department of Hematology, University Hospitals Leuven, 3000 Leuven, Belgium; 3Laboratoire de Parasitologie-Mycologie, AP-HP, Groupe Hospitalier Saint-Louis-Lariboisière-Fernand-Widal, 75010 Paris, France; nesrine.aissaoui@aphp.fr (N.A.); maud.gits-muselli@aphp.fr (M.G.-M.); samia.hamane@aphp.fr (S.H.); stephane.bretagne@pasteur.fr (S.B.); 4Department of Internal Medicine, Section of Infectious Diseases and Tropical Medicine, Medical University of Graz, 8036 Graz, Austria; juergen.prattes@medunigraz.at; 5Diagnostic & Research Institute of Hygiene, Microbiology and Environmental Medicine, Medical University of Graz, 8036 Graz, Austria; harald.kessler@medunigraz.at; 6Department of Clinical and Molecular Microbiology, University Hospital Centre Zagreb, School of Medicine, University of Zagreb, 10000 Zagreb, Croatia; imarekov@kbc-zagreb.hr (I.M.); sanja.plesko@kbc-zagreb.hr (S.P.); 7Institute for Clinical Hygiene, Medical Microbiology and Clinical Infectiology, Paracelsus Medical University, Nuremberg Hospital, 90419 Nuremberg, Germany; Joerg.Steinmann@klinikum-nuernberg.de; 8Institute of Medical Microbiology, University Hospital Essen, 45122 Essen, Germany; Ulrike.Scharmann@uk-essen.de; 9Department of Laboratory Medicine and National Reference Center for Mycosis, University Hospitals Leuven, 3000 Leuven, Belgium; 10Service de Maladies Infectieuses et Tropicales, AP-HP, Groupe Hospitalier Saint-Louis-Lariboisière-Fernand-Widal, 75010 Paris, France; blandine.denis@aphp.fr; 11Department of Infectious Agents, Université de Paris, 75006 Paris, France; 12Molecular Mycology Unit, Centre National de la Recherche Scientifique (CNRS), Unité Mixte de Recherche UMR2000, Centre National de Référence Mycoses Invasives et Antifongiques (CNRMA), Institut Pasteur, 75724 Paris, France

**Keywords:** *Pneumocystis jirovecii*, qPCR, broncho-alveolar lavage fluid, fungal load, biomarker, (1,3)-β-d-glucan, non-HIV patient

## Abstract

*Pneumocystis jirovecii* pneumonia is a difficult invasive infection to diagnose. Apart from microscopy of respiratory specimens, two diagnostic tests are increasingly used including real-time quantitative PCR (qPCR) of respiratory specimens, mainly in bronchoalveolar lavage fluids (BAL), and serum β-1,3-d-glucan (BDG). It is still unclear how these two biomarkers can be used and interpreted in various patient populations. Here we analyzed retrospectively and multicentrically the correlation between BAL qPCR and serum BDG in various patient population, including mainly non-HIV patients. It appeared that a good correlation can be obtained in HIV patients and solid organ transplant recipients but no correlation can be observed in patients with hematologic malignancies, solid cancer, and systemic diseases. This observation reinforces recent data suggesting that BDG is not the best marker of PCP in non-HIV patients, with potential false positives due to other IFI or bacterial infections and false-negatives due to low fungal load and low BDG release.

## 1. Introduction

*Pneumocystis jirovecii* pneumonia (PCP) is one of the most prevalent invasive fungal infections [1]. It is still one of the main infections revealing AIDS in western countries [2]. In parallel, it is now mainly diagnosed in non-HIV patients, such as patients treated for hematological malignancies, solid organ transplant recipients, or patients treated with immunosuppressive therapies [3]. The clinical presentation and biological features associated with PCP are different in HIV and non-HIV patients, suggesting that the disease has a different pathophysiology with a more acute disease presentation and an increased mortality in non-HIV patients [4].

The diagnosis of PCP has relied on microscopy with immunofluorescence as the most sensitive test to visualize the trophic forms and asci containing ascospores in respiratory specimens since the 1970s [5]. Since then, PCR for the sensitive detection of *P. jirovecii* DNA in respiratory specimens [6] and 1,3-β-d-glucan (BDG) assays for the detection of specific polysaccharides from the asci cell wall in serum [7]. were developed and evaluated in the 1990s. More recently, real-time quantitative PCR (qPCR) assays have been developed and are nowadays the only PCR method recommended for diagnosis [8]. Recommendations for diagnosis of PCP in patients with hematological malignancies rely on qPCR and BDG testing, with various flow-chart procedures depending on the type of specimen undergoing testing [8]. Evaluation of qPCR assays have shown excellent sensitivity and negative predictive values, allowing qPCR in bronchoalveolar lavage (BAL) fluid to be used to rule out the infection [8,9]. On the other hand, serum BDG initially showed excellent performance with 94.8% sensitivity and an area under the curve (AUC) of 0.965 [10]. However, recent studies, including another meta-analysis, tempered these results and highlighted the fact that the diagnostic accuracy of serum BDG in non-HIV patients seemed to be less sensitive [11,12]. The meta-analysis from Corpo et al. concluded that pooled sensitivity of BDG is thus insufficient to exclude PCP [11]. One strategy could therefore be to combine BDG in serum and qPCR in respiratory specimens to obtain more accurate results. When plotting BAL fungal load by qPCR and serum BDG titers taken within 15 days of bronchoscopy (*n* = 46), a weak but significant correlation between the log-transformed values was found with a regression line significantly different of 0 (*p* = 0.0005) and a slope of 0.33 (*R*^2^ = 0.2459) [13]. Another study based on cancer patient specimens also reported a weak correlation between both markers (Spearman index at 0.38) [12].

The aim of this study was to investigate the potential correlation between BAL qPCR fungal load and serum BDG in various populations of patients based on their underlying diseases.

## 2. Materials and Methods

### 2.1. Participating Centers and Patients

All Excellence Centers from the ECMM network (https://www.ecmm.info/ecmm-excellence-centers/) were invited to participate to the study and five centers accepted to include patients in this multicenter study. Centers were anonymized as Center 1 to Center 5.

Patients were retrospectively enrolled between 1 January 2015 to 31 December 2019. Inclusion of the cases required that BAL fluid was tested with qPCR and a minimum of one serum for BDG was tested within a time frame of 7 days before or after BAL sampling. Coinfection with other invasive fungal infections were collected in parallel.

We anonymously collected the underlying disease and the final diagnosis, classified by the treating clinician as PCP or *Pneumocystis* carriage (PCC). PCP was defined as cases associated with a positive *Pneumocystis jirovecii* qPCR in BAL sample together with classical radiological signs (bilateral ground glass opacities on chest computed tomography or bilateral diffuse interstitial infiltrates on chest X-ray), and any of the following clinical signs and symptoms: dyspnea, cough, or hypoxemia.

### 2.2. (1,3)-β-d-Glucan Assay

BDG was performed in each center using the Fungitell assay (Associates of Cape Cod) in accordance with the manufacturer’s instructions. BDG > 80 pg/mL was considered positive. BDG categories were defined as <80, 80–200, 200–523, and >523 pg/mL to separate negative, weakly positive, positive, and strongly positive results. BDG > 523 pg/mL were arbitrarily set at 524 pg/mL.

### 2.3. qPCR Assays and Calibration

qPCR was performed in each center using various kits and assays (Appendix A). A calibrator specimen (pooled *P. jirovecii*-positive DNA in pooled *P. jirovecii*-negative DNA extracted from BAL as described earlier [14]) was sent to all participating centers and tested using the local routine qPCR assay. The quantification cycle (Cq) value of the calibrator was then used to adjust all previously collected qPCR Cq values from all respective centers. Basically, the mean Cq value of the calibrator obtained from the five centers was calculated. An adjustment factor was calculated by subtracting the mean Cq value of the calibrator to the Cq value of the calibrator obtained by each center. The adjustment factor of each center was then subtracted from the Cq values of each BAL tested in that specific center in order to obtain the adjusted-Cq value.

### 2.4. Graphs and Statistical Analysis

Graph and statistical analyses were performed using Prism v. 8.4.3 (GraphPad Software). ANOVA Kruskal–Wallis tests were performed on categorical data and Mann–Whitney unpaired non-parametric tests were performed accordingly. Cq values and BDG titers (pg/mL) were used for qPCR and BDG data analysis. Medians and interquartile ranges are provided for data with non-gaussian distributions.

## 3. Results

A total of 147 patients were enrolled in this study, including 117 cases of Pneumocystis pneumonia (PCP) and 30 cases of Pneumocystis carriage (PCC). The distribution of the enrolment is shown in Table 1. Most of the patients were HIV-negative (91.2%), including hematological malignancies (46.3%), solid cancer (17.7%), solid organ transplant (8.8%), systemic diseases (9.5%), and other underlying diseases (8.8%) (Table 2). The median delay between serum sampling and BAL was 0 days [IQR 0–1].

The fungal load was significantly different between PCP and PCC patients with a median Cq of 28.0 [IQR 25.9–30.9] in PCP vs. 35.0 [IQR 33.5–36.8] in PCC patients (*p* < 0.0001) (Figure 1A). BDG was also significantly different between PCP and PCC patients with a median titer of 452 pg/mL [IQR 158–524] in PCP vs. 16.30 [IQR 7.70–79.25] (*p* < 0.0001) (Figure 1B). The AUC was 0.922 for qPCR in BAL, and 0.928 for serum BDG. The optimal threshold values to discriminate between PCP and PCC appeared to be Cq = 34 (sensitivity, 70%; specificity, 95.7%; positive likelihood ratio, 16.4) and BDG = 192 pg/mL (sensitivity, 100%; specificity, 72.6%; positive likelihood ratio, 3.6) (Figure 1C,D, Appendix A).

In the full cohort, the regression line between fungal load (Cq value) and BDG titer was significantly different from 0 (*p* < 0.0001) and showed a *R*^2^ of 0.17 (Figure 2A). The distribution of the BAL fungal load was significantly different in the following BDG categories < 80, 80–200, 200–523, >523 pg/mL (Anova *p* < 0.0001), with significantly lower Cq values in patients with >523 pg/mL compared to other categories (Figure 2B).

However, when focusing on various underlying diseases, some differences in the correlation appeared with the highest *R*^2^ at 0.681 in HIV patients followed by solid organ transplant recipients (*R*^2^ = 0.573), systemic diseases (*R*^2^ = 0.326), and other underlying diseases (*R*^2^ = 0.179). A very weak correlation was observed in patients with hematological malignancies (*R*^2^ = 0.097) and solid cancer (*R*^2^ = 0.078) (Figure 3).

The distribution of the BAL Cq values was statistically different between the four BDG groups in hematological malignancies, solid cancer, solid organ transplant recipients, and others (*p* < 0.001). In all four groups, patients with BDG > 523 had significantly lower Cq values than patients with lower BDG values (Figure 4). Of note, in HIV and systemic disease patients, Cq values were not significantly different between the different BDG categories.

## 4. Discussion

In this multicenter international retrospective study, we found a weak correlation between fungal load in BAL (as estimated by qPCR Cq values) and serum BDG value. Previous studies suggested that BDG in serum could be used as an estimator of the fungal burden, albeit with a weak correlation [12,13]. One possible explanation for the weak correlation could be the delay between BAL sampling and serum BDG testing, up to 15 days in one study [13], with no simultaneous increase or decrease, specifically after treatment, between the fungal load and the circulating antigen, as seen in *Candida* infections [15]. Indeed, we found a significant difference in fungal burden between patients having the highest BDG levels and the remainders. However, despite having most of the serum samples on the day of BAL sampling in this study, or the day before/after BAL sampling, we also found only a weak correlation with an *R*^2^ of 0.17 for the whole cohort.

To analyze the correlation between qPCR and BDG, we faced the problem of heterogeneity of qPCR methods used in the five participating centers. This was not the case with BDG, since all participants used the Fungitell kit. Therefore, we circumvented this problem by using a qPCR calibrator sample sent to all participants. This allowed us to observe the differences in terms of quantification for a given sample in the various centers, as shown previously [14], and to calculate the adjusted qPCR result.

Since the diagnostic performance of BDG is different between HIV and non-HIV patients [11,12], we can presume a role for the immune system in the process of clearance from and/or release of BDG into the circulation. This is also evidenced by the differences in pathogenesis between these groups [4]. For this reason, we also stratified patients based on the underlying disease, and found a disease-dependent correlation between BDG and fungal load. In agreement with recent meta-analysis results, the strongest correlation was found in HIV-patients (*R*^2^ = 0.573), with the weakest correlation in solid cancer patients (*R*^2^ = 0.078) and hematologic malignancies (*R*^2^ = 0.097).

Other possible sources of increased variability–and thus a lower *R*^2^ in cancer and hematology patients include concomitant invasive fungal infections (IFI) which could make BDG testing positive. In our study, only three non-HIV patients were reported to have a concomitant IFI (three invasive aspergilloses) together with PCP. One had a negative and two had positive (>500 pg/mL) BDG tests. Indeed, this cannot explain why BDG and fungal load do not correlate well in non-HIV patients. One additional explanation could be the increased rate of concomitant bacterial infection which can lead to false positive BDG results [16,17,18]. In addition, the Fungitell assay is known to have significant analytical variability [19], which could impact the value of the BDG titer. On the other hand, the method of collection of BAL is known to be variable from a center to another and even from different wards (ICU vs. pneumology), allowing the introduction of variability in the quantification of the fungal load by qPCR [20]. Moreover, although there is a good correlation between Cq values and microscopic fungal burden (trophic or cystic forms per optical field) [21,22], this correlation is not perfect, adding another layer of variability [23].

It appears more and more clearly that in hematology, cancer, and systemic disease patients, BDG cannot rule out PCP. A positive BDG in this population should prompt a full microbiological work up to allow bacterial and fungal diagnosis outside PCP. This is why ECIL-5 experts placed BAL and *Pneumocystis jirovecii* detection as the first test to be performed in these patient populations [8] with BDG testing done as a second step. In contrast, BDG testing in addition to induced sputum for the detection of *Pneumocystis* seems to be a more suitable strategy to diagnose PCP in HIV patients. In non-HIV patients, *Pneumocystis jirovecii* qPCR plays a central role and is being standardized in order to implement it in clinical studies on the performance of a standard qPCR on the diagnosis of PCP.

Our study has several limitations. The study was not designed to evaluate the performance of BDG or qPCR, since the clinicians were aware of the results when they classified the patients. Indeed, some clinicians have integrated that some PCP can be BDG negative. It would have been interesting to know whether some BDG positive/qPCR negative patients were classified as PCP, but this will be investigated in the future.

We were able to enroll only a limited number of HIV-patients which prevents having strong evidence of the better correlation between BAL qPCR and serum BDG in HIV patients. However, comparing the results in HIV and in the other underlying diseases categories are very interesting and could give rise to specific studies in the future.

Another limitation is that, although we included samples from five expertise centers, the majority was included in two centers, which could result in confounding by a center effect. However, given that all parameters in this study are diagnostic parameters, any potential center-related differences in clinical management will not affect the (diagnostic) parameters used in this study. We therefore believe the potential bias to be small.

Additionally, the use of different qPCR methods between the different centers could be a potential source of variability. Although we did correct for this using a calibrator-based standardization method, some remaining variability cannot be excluded. Additionally, differences in BAL sampling such as the volume of liquid injected, can be a further source of variability. Moreover, the sample sizes of certain disease subgroups (such as HIV) were small, making statistical analysis difficult.

In conclusion, serum BDG values appear to offer some estimation of the fungal burden depending on the underlying disease (i.e., especially in HIV patients), but are not a perfect predictor of the qPCR results.

## Figures and Tables

**Figure 1 jof-06-00327-f001:**
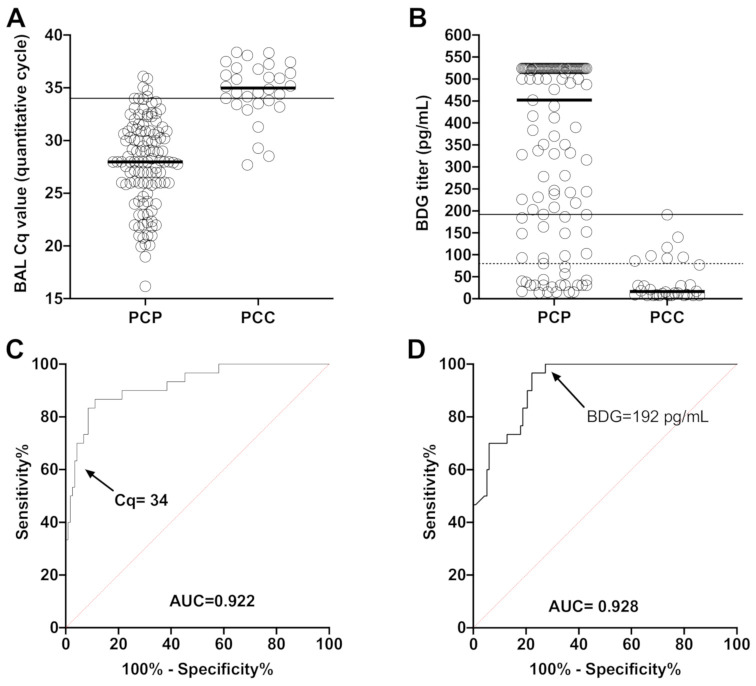
Distribution of the BAL Cq values (**A**) and BDG titers (**B**) and corresponding ROC curves for BAL Cq values (**C**) and BDG titers (**D**) comparing *Pneumocystis* pneumonia (PCP) and *Pneumocystis* carriage patients (PCC). Solid lines represent calculated optimal thresholds and black/red dotted lines the manufacturer threshold of the assay (80 pg/mL for Fungitell BDG assay).

**Figure 2 jof-06-00327-f002:**
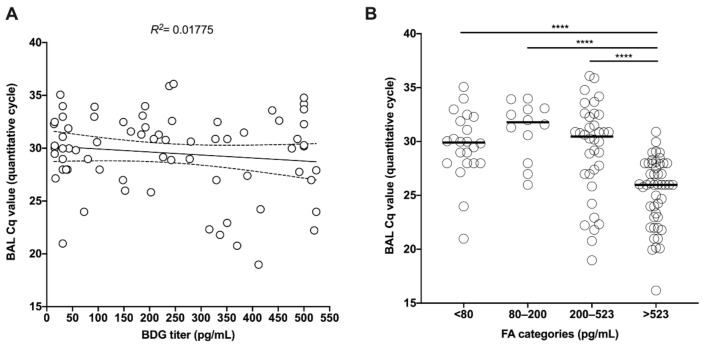
Linear (**A**) and categorial (**B**) correlation between BAL Cq value and BDG titer (pg/mL) in PCP patients. **** *p* < 0.0001.

**Figure 3 jof-06-00327-f003:**
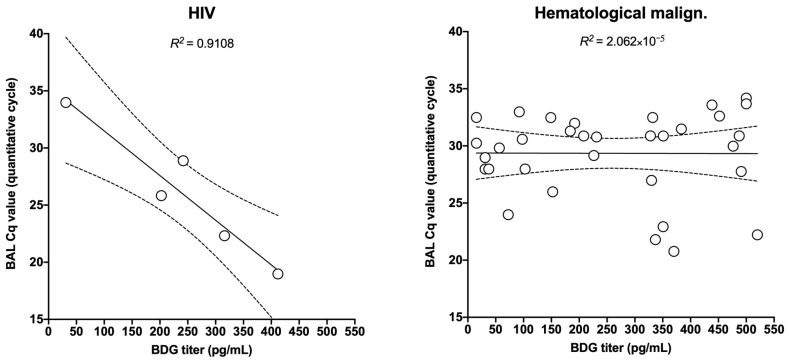
Linear correlation between BAL Cq value and BDG titer (pg/mL) according to the underlying diseases.

**Figure 4 jof-06-00327-f004:**
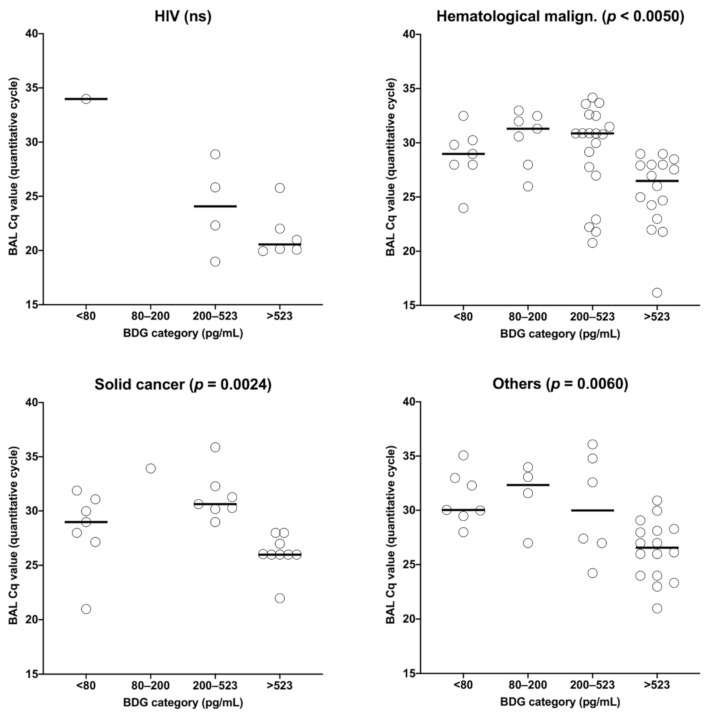
Categorial correlation between BAL Cq value and BDG titer (pg/mL) according to the underlying diseases.

**Table 1 jof-06-00327-t001:** Distribution of enrolment in the five centers.

	Patients Enrolled, *n* (%)	PCP, *n* (%)	PCC, *n* (%)
Center 1	66 (44.9)	66 (56.4)	0
Center 2	65 (44.2)	37 (31.6)	28 (93.3)
Center 3	8 (5.4)	6 (5.1)	2 (6.7)
Center 4	4 (2.7)	4 (3.4)	0
Center 5	4 (2.7)	4 (3.4)	0
Total	147	117	30

**Table 2 jof-06-00327-t002:** Distribution of the underlying diseases.

Underlying Diseases	All, *n* (%)	PCP, *n* (%)	PCC, *n* (%)
Hematological malignancies	68 (46.3)	49 (41.9)	19 (63.3)
Solid Cancer	26 (17.7)	24 (20.5)	2 (6.7)
HIV	13 (8.8)	11 (9.4)	2 (6.7)
Solid organ transplantation	13 (8.8)	11 (9.4)	2 (6.7)
Systemic disease	14 (9.5)	10 (8.5)	4 (13.3)
Other	13 (8.8)	12 (10.3)	1 (3.3)
Total	147	117	30

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
