# Peer review of "Variable Correlation between Bronchoalveolar Lavage Fluid Fungal Load and Serum-(1,3)-β-d-Glucan in Patients with Pneumocystosis—A Multicenter ECMM Excellence Center Study"

_jof, 2020, doi:10.3390/jof6040327_

Round 1
Reviewer 1 Report
The present multi-center study by Mercier et al., evaluates the correlation between BAL qPCR and serum BDG in various patient’s population including mainly non-HIV patients. The authors conclude that BDG is not the best marker of PCP in non-HIV patients, with potential false positive due to other IFI or bacterial infections and false-negative due to low fungal load and low BDG release. Although the study objective is clinically relevant, the study has several major limitations and it is very difficult (if not impossible) to reach the present conclusion with the present data.
Major comments:
- The overall sample size is rather small with less than 150 patients.
- The study includes data from five centers with considerably different sample size. Two centers have each of 8 patients only. Therefore, there is a significant bias with two centers delivering 131/147 patients.
- The authors claim that there is a difference in the performance of BDG in patients with HIV and non-HIV. However, the HIV group consists of 13 patients (8.8% of the total). It is not possible to draw any conclusion with such a small sample size. Please exclude.
- Similarly as above Fig 3 and 4 analyzes small groups of patients acc to the BAL Cq value and BDG titers. I suggest that the authors group the non-HIV patients in three groups as Hematological malignancies, Solid Cancer and others.
- The authors claim that IFI has an impact on BDG positivity as reported previously. It is also known that the patients with mild infections and colonization may have BDG positive results. Did the authors investigate if the patients were colonized with e.g. Candida spp.?
Minor comments:
- No
Author Response
Reviewer 1 :
The present multi-center study by Mercier et al., evaluates the correlation between BAL qPCR and serum BDG in various patient’s population including mainly non-HIV patients. The authors conclude that BDG is not the best marker of PCP in non-HIV patients, with potential false positive due to other IFI or bacterial infections and false-negative due to low fungal load and low BDG release. Although the study objective is clinically relevant, the study has several major limitations and it is very difficult (if not impossible) to reach the present conclusion with the present data.
Major comments:
The overall sample size is rather small with less than 150 patients.
We agree with the reviewer that a larger sample size would have been beneficial and would have allowed us to make more robust deductions on the relation between BDG, qPCR and underlying disease. However, as the reviewer remarks in comment nr 2, this study already incorporates a large number of fungal expertise centers from the ECMM network. Even despite this collaboration of specialized centers, a larger sample size was not attainable. Furthermore, our study is larger than or equally sized to other studies in the field (see for example references 12 and 13).
The study includes data from five centers with considerably different sample size. Two centers have each of 8 patients only. Therefore, there is a significant bias with two centers delivering 131/147 patients.
We certainly agree that a center effect is a possible confounder, and have added this to the limitations section of the paper. However, given that all parameters are diagnostic parameters, any potential center-related differences in clinical management will not affect the (diagnostic) parameters used in this study. We also limited the potential variation of the qPCR methods by applying a normalization factor based on the results of a calibrator panel specimen tested in all centers (see M&M section). We therefore believe the potential bias to be small to non-existent. (p. 14, lines 232-236).
The authors claim that there is a difference in the performance of BDG in patients with HIV and non-HIV. However, the HIV group consists of 13 patients (8.8% of the total). It is not possible to draw any conclusion with such a small sample size. Please exclude.
We acknowledge that we did not enroll a lot of HIV-patients in our study since the protocol was not oriented towards a specific patient population. In practice, HIV-patients were not often tested simultaneously by both qPCR in BAL and BDG in serum. Therefore, only 13 patients were enrolled and of these 13 (11 only had PCP), 6 had a BDG >523. These 6 were thus excluded of the correlation analysis in PCP partients (Figure 3), allowing analysis only on 5 patients with PCP. Indeed, HIV patients are known to have a higher fungal load and so higher BDG titers at diagnosis (>523). This is a result per se which is illustrated in Figure 4.
Indeed, in our center, BAL is done in second intention after an induced sputum is tested with a qPCR. In HIV patients induced sputum is really efficient to recover P. jirovecii in HIV since the fungal load is high in these patients and so a non-invasive specimen is enough to show a high burden in this specific high -risk population.
So, from our centers it is quite clear that qPCR in BAL would not be obtained frequently. This explains why only a few HIV patients have been enrolled in our study.
However, we think that it is relevant to show the correlation in HIV patients in comparison other patient populations. However, to better clarify the fact that conclusions are difficult to be obtained with HIV-patients due to the low number of patients enrolled, we discuss this limitation in a dedicated paragraph (p. 14, lines 228-231).
Similarly as above Fig 3 and 4 analyzes small groups of patients acc to the BAL Cq value and BDG titers. I suggest that the authors group the non-HIV patients in three groups as Hematological malignancies, Solid Cancer and others.
We thank Reviewer 1 for this comment and have modified figures 3 and 4 accordingly (see new figures 3 and 4).
The authors claim that IFI has an impact on BDG positivity as reported previously. It is also known that the patients with mild infections and colonization may have BDG positive results. Did the authors investigate if the patients were colonized with e.g. Candida spp.?
We did collect data on concomitant bacterial, viral or fungal infections. Data on other fungal colonizers was not collected. See p. 13, lines 199-205. Candida colonization of the airways may occur in more than 50% of hospitalized patients having a respiratory specimen tested in our laboratory. Since BDG is not performed in BAL but in serum, we do not think that serum BDG would be affected by airway Candida colonization, otherwise 50% of the patients would have a positive BDG which has never been shown in any studies dealing with BDG testing in serum. However, we agree that major colonization of the digestive tract in addition to severe mucositis could be a cause of false positivity of BDG, albeit difficult to prove.
Reviewer 2 Report
- Lines 57-59. The dots after all references, even if the sentence has not finished, makes it a little hard to read.
- Table 1: ‘patients enrolled’ à Please use capital P
- Line 87: This sentence is hard to read. Please rephrase.
- Line 118: is shown IN Table 1.
- Table 2: The title does not make sense. Maybe it would be better like this: Distribution of the underlying patients’ diseases.
- Line 154: ‘with in all cases the >523 group harboring significantly lower Cq values than the other groups’. Not clear. Please rephrase.
- Some groups, like the one with the Hiv patients had very few patients. Thus, statistical analysis with them could not be quite representative (eg Fig 4). This should be stated in the limitations section.
- Line 185: with recent meta-ANALYSIS
- Line 203: Fragmented sentence?
- This study lacks a limitation section. I would expect to see one mentioning the abovementioned limitation in statistics in the groups with the few patients, the heterogeneity of qPCR performance among the centers, even though a method was occupied to make the results more comparable, the possible heterogeneity in the acquisition of the sample where the qPCR was performed (sputum?, BAL?) etc.
- A table, probably in the supplementary material, could be added, showing the conditions of the patients with ‘other’ conditions.
- Line 211: I don’t understand what the authors mean at the conclusion section when they say ‘’ depending on the underlying disease’’. Do they refer to the HIV patients? Because they did not mention something about another disease in the HIV population.
Author Response
Lines 57-59. The dots after all references, even if the sentence has not finished, makes it a little hard to read.
The layout has been adjusted.
Table 1: ‘patients enrolled’ à Please use capital P
This has been adjusted
Line 87: This sentence is hard to read. Please rephrase.
We have rephrased this line (p. 3, line 92-94).
Line 118: is shown IN Table 1.
Thank you for noticing this, we have added the word “in”
Table 2: The title does not make sense. Maybe it would be better like this: Distribution of the underlying patients’ diseases.
We have removed the word “patients” to make the title clearer
Line 154: ‘with in all cases the >523 group harboring significantly lower Cq values than the other groups’. Not clear. Please rephrase.
We have rephrased this sentence to make it easier to read (p. 9, lines 160-163)
Some groups, like the one with the Hiv patients had very few patients. Thus, statistical analysis with them could not be quite representative (eg Fig 4). This should be stated in the limitations section.
We have now added this to the limitations section (p. 14, lines 240-242)
Line 185: with recent meta-ANALYSIS
Thank you for noticing this, we have updated the sentence.
Line 203: Fragmented sentence?
Indeed, we have adjusted this.
This study lacks a limitation section. I would expect to see one mentioning the abovementioned limitation in statistics in the groups with the few patients, the heterogeneity of qPCR performance among the centers, even though a method was occupied to make the results more comparable, the possible heterogeneity in the acquisition of the sample where the qPCR was performed (sputum?, BAL?) etc.
We have added a section on the limitations of our study (p. 14-15, lines 223-242)
A table, probably in the supplementary material, could be added, showing the conditions of the patients with ‘other’ conditions.
As asked by reviewer 1, we pooled systemic diseases, SOT and Others in the category “Others” since the number of patients was considered as low in each group. Therefore, “others” now includes systemic diseases, SOT, and patients with other rare cause of immunosuppression that do not fall into the other categories (for ex. chronic inflammatory bowel disorders).
Line 211: I don’t understand what the authors mean at the conclusion section when they say ‘’ depending on the underlying disease’’. Do they refer to the HIV patients? Because they did not mention something about another disease in the HIV population.
Indeed! We have rephrased this sentence. (p. 14, lines 244-246)
Round 2
Reviewer 1 Report
I would like to thank the authors for replying all my comments in detail.